# Disease burden of asbestos-related diseases in China (1990–2023) based on GBD estimates: A call for stronger labor protection laws

Duo An[1], Zherui Shi[2], Chenxiao Zhang[1], Shichang Song[3], Lu Wang[3], Hanchao Li[4]*

**1** School of Law, Xi'an Jiaotong University, Xi'an, Shaanxi Province, the People's Republic of China,
**2** Law School, XinJiang University, Urumqi City, Xinjiang Province, the People's Republic of China,
**3** School of Public Health, Xi'an Jiaotong University Health Science Center, Xi'an, Shaanxi Province, the People's Republic of China, **4** Department of Rheumatology, the First Affiliated Hospital of Xi'an Jiaotong University, Xi'an, Shaanxi Province, the People's Republic of China

* lihch1991@xjtu.edu.cn

## Abstract

### Background

Asbestos exposure remains a persistent occupational hazard in China, yet updated national estimates of asbestos-related diseases (ARDs) after 2019 are scarce. This study quantified long-term trends and demographic patterns of ARDs from 1990 to 2023 using Global Burden of Disease data and joinpoint regression.

### Methods

We analyzed incidence, prevalence, deaths, and disability-adjusted life years (DALYs) for asbestosis and asbestos-attributable cancers (mesothelioma, tracheal/bronchus/lung cancer, laryngeal cancer, and ovarian cancer). Absolute numbers and age-standardized rates were assessed overall and stratified by sex and age. joinpoint regression identified significant temporal inflection points.

### Results

The absolute burden of ARDs increased continuously from 1990 to 2023. Age-standardized prevalence and incidence rates of asbestosis peaked in 2001, while mortality and DALY rates peaked in 2004. Major turning points for asbestos-attributable cancers occurred around 2010–2011, marking historical peaks followed by declines. A modeled increase in mortality and DALYs was observed from 2020 to 2022 across nearly all ARDs. Males consistently demonstrated higher burdens than females, and older adults (≥65 years) carried the greatest burden, with a secondary mesothelioma peak at 55–59 years in males.

**Data availability statement:** The data and R scripts that support the findings of this study have been deposited in the Figshare repository and are publicly available at https://doi.org/10.6084/m9.figshare.31745317. All relevant data are included in this published article and its supplementary information files or are available from the repository.

**Funding:** The author(s) received no specific funding for this work.

**Competing interests:** The authors declare they have no competing interests.

## Conclusions

Although ARD indicators have declined from historical peaks, a statistically modeled increase was observed in 2020–2022, warranting continued public-health attention. These findings aim to provide evidence for clinicians, epidemiologists, and policymakers to strengthen occupational disease prevention, reinforce labor protection laws, and improve asbestos-control policies in China.

## Introduction

Asbestos-related diseases (ARDs) comprise a group of malignant and non-malignant conditions caused by inhalation of asbestos fibers [1]. Asbestosis, a progressive interstitial lung disease driven by the high biopersistence of inhaled fibers, represents the most typical non-malignant ARD [2]. Chronic exposure can lead to persistent pulmonary inflammation, fibrosis, and irreversible lung injury. In addition, asbestos exposure is a well-established cause of several cancers, including mesothelioma, lung cancer, laryngeal cancer, and ovarian cancer [3]. Despite asbestos bans in more than 60 countries [4], the global burden of ARDs continues to rise, particularly in low- and middle-income countries [5,6].

China has long been one of the world's major producers and consumers of asbestos, and millions of workers have experienced occupational exposure [7]. National production has not been fully banned, and the number of asbestos-related enterprises continued to increase at least until 2019 [7]. Asbestos exposure is the principal cause of malignant mesothelioma, and in China, it is also an important risk factor for laryngeal cancer [8], ovarian cancer [9], and occupationally related lung cancer [10]. Despite this, comprehensive assessments of the disease burden of both asbestosis and asbestos-attributable cancers in China are still lacking.

The Global Burden of Disease (GBD) study provides a standardized framework for evaluating the burden of diseases and risk factors across countries and over time. Although earlier GBD-based studies have described selected ARDs in China up to 2019, no study has provided an integrated and up-to-date evaluation of asbestosis together with asbestos-attributable cancers through 2023. Updated analyses are essential for understanding current patterns, guiding prevention strategies, and informing occupational health policy.

Therefore, this study used GBD 1990–2023 data to quantify long-term trends in the burden of asbestosis and four asbestos-attributable cancers in China. We assessed incidence, prevalence, deaths, and disability-adjusted life years (DALYs), examined age-standardized rates, evaluated temporal trends using joinpoint regression, and analyzed demographic patterns by sex and age. These findings are intended to serve as descriptive evidence that may inform clinicians, epidemiologists, and policymakers in efforts to strengthen occupational disease prevention, labor protection laws, and asbestos-control policies in China.

## Methods

### Data source

Data were obtained from the Global Burden of Disease (GBD) 1990–2023 study, a comprehensive framework for estimating disease burden across countries and over time [11]. Information on asbestosis and asbestos-attributable cancers was extracted through the Global Health Data Exchange (GHDx) GBD Results Tool (https://vizhub.healthdata.org/gbd-results/), with the last access date being July 29, 2025. The GBD study integrates data from vital registration systems, cancer registries, surveys, and modeling strategies to produce comparable estimates of incidence, prevalence, mortality, and disability-adjusted life years (DALYs) for all countries.

In the GBD framework, the burden of cancers attributable to asbestos exposure is estimated using the comparative risk assessment approach. This framework combines exposure distributions, relative risk estimates from epidemiological studies, and theoretical minimum risk exposure levels to calculate population attributable fractions (PAFs) [12]. The PAFs are then applied to total disease burden estimates to derive asbestos-attributable deaths and DALYs for each cancer type.

### Measures and data extraction

For asbestosis, we extracted annual estimates of incidence, prevalence, deaths, and DALYs, including both absolute numbers and age-standardized rates.

For asbestos-attributable cancers, we obtained deaths and DALYs attributable specifically to the risk factor "asbestos exposure." Consistent with the GBD framework, these include mesothelioma, tracheal/bronchus/lung cancer, laryngeal cancer, and ovarian cancer. Ovarian cancer is included as an asbestos-attributable cancer in accordance with the Organization (WHO) and the International Labour Organization (ILO) criteria, which recognize sufficient epidemiological evidence for a causal relationship between occupational asbestos exposure and ovarian cancer [13]. The inclusion of ovarian cancer alongside the other three cancer types reflects the current evidence base used in GBD risk attribution.

All data were retrieved for China and stratified by sex and 5-year age groups. Age-standardized rates were calculated by the GBD study using the GBD standard population, enabling comparison across years independent of demographic changes.

### Data visualization

Indicators were analyzed both as absolute numbers and as age-standardized rates per 100,000 population. Data cleaning and restructuring were performed in R software (version 4.2.3) using the *dplyr* and *reshape2* packages. Figures were generated using *ggplot2* to visualize temporal patterns and demographic differences.

### Joinpoint regression analysis

Temporal trends from 1990 to 2023 were evaluated using joinpoint regression (joinpoint Regression Program version 5.2.0.0, National Cancer Institute, USA) [14]. Log-linear models were applied to identify points where trends changed significantly ("joinpoints").

GBD age-standardized rates (point estimates) served as the dependent variable, with corresponding standard errors incorporated as weights (Heteroscedastic/Correlated Errors Option: "Standard Error (Provided)," uncorrelated errors model) to account for the varying precision of each observation. The maximum number of joinpoints was set to six, and the final model was selected using Weighted Bayesian Information Criterion (BIC).

For each trend segment, we calculated the annual percent change (APC) with 95% confidence intervals. The average annual percent change (AAPC) summarized overall trends during the study period. A trend was considered statistically significant when the APC or AAPC differed from zero at $p < 0.05$.

## Ethics approval and consent to participate

This study used only publicly available de-identified data from the GBD database. Ethical approval was reviewed and approved by the Ethics Committee of the First Affiliated Hospital of Xi'an Jiaotong University, which confirmed that informed consent was not required for the use of secondary data.

## Results

### Evolution of asbestosis burden in China, 1990–2023

From 1990 to 2022, the absolute numbers of asbestosis prevalence, incidence, deaths, and DALYs increased steadily, followed by a slight decline in 2023 (Fig 1). Joinpoint regression (Fig 2) showed highly consistent temporal patterns between age-standardized prevalence (ASPRs) and incidence rates (ASIRs), both of which rose rapidly during 1990–1994, increased more slowly until 2000, and then reached their historical peaks before entering a prolonged decline (S1 Table). Age-standardized mortality (ASMRs) and DALY rates (ASDRs) also shared similar patterns. ASMRs remained stable from 1990 to 1994, increased sharply during 1994–2004, and then declined through 2015. ASDRs increased continuously from 1990 to 2004. Both ASMRs and ASDRs plateaued during 2015–2019 and increased again after 2019 ($p < 0.05$).

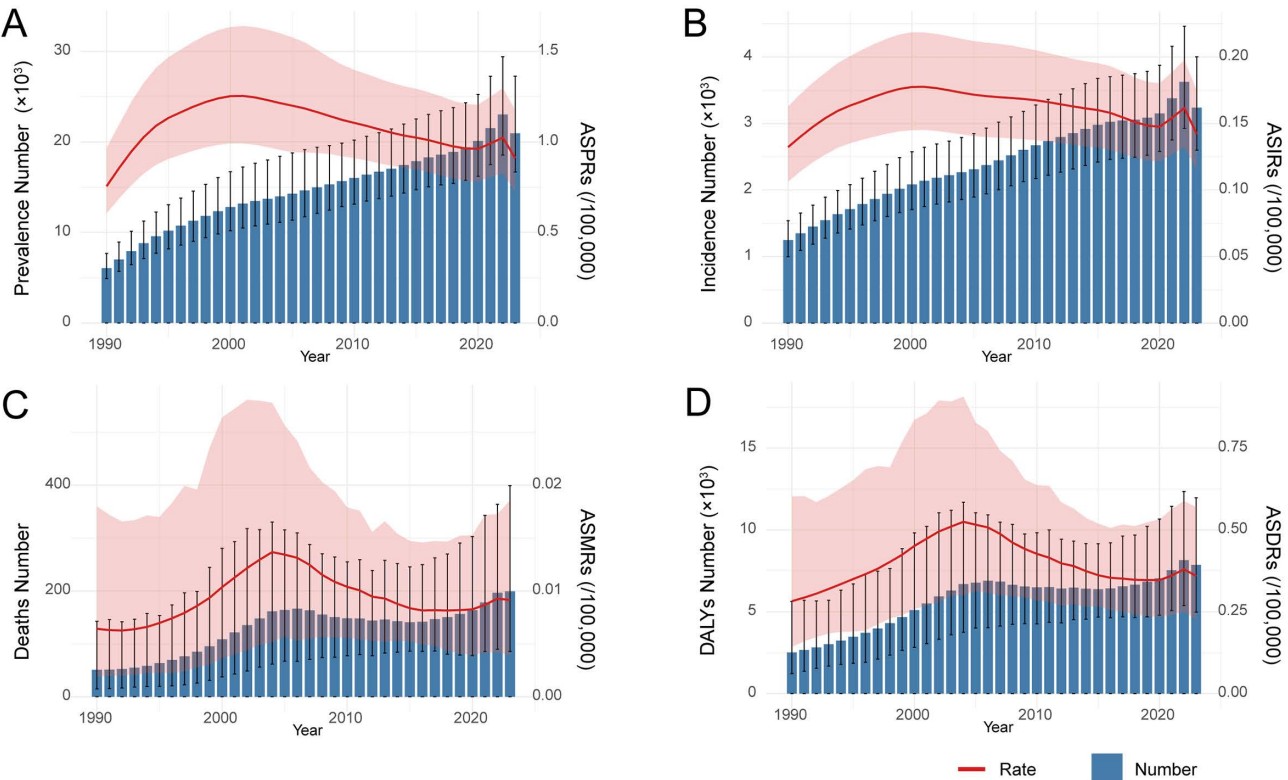

**Fig 1. Disease burden of asbestosis in China.** The number and age-standardized rates of prevalence **(A)**, incidence **(B)**, deaths **(C)**, and DALYs **(D)** for asbestosis. Abbreviations: ASPRs: age-standardized prevalence rates; ASIRs: age-standardized incidence rates; ASMRs: age-standardized mortality rates; DALYs: disability-adjusted life years; ASDRs: age-standardized DALY rates;.

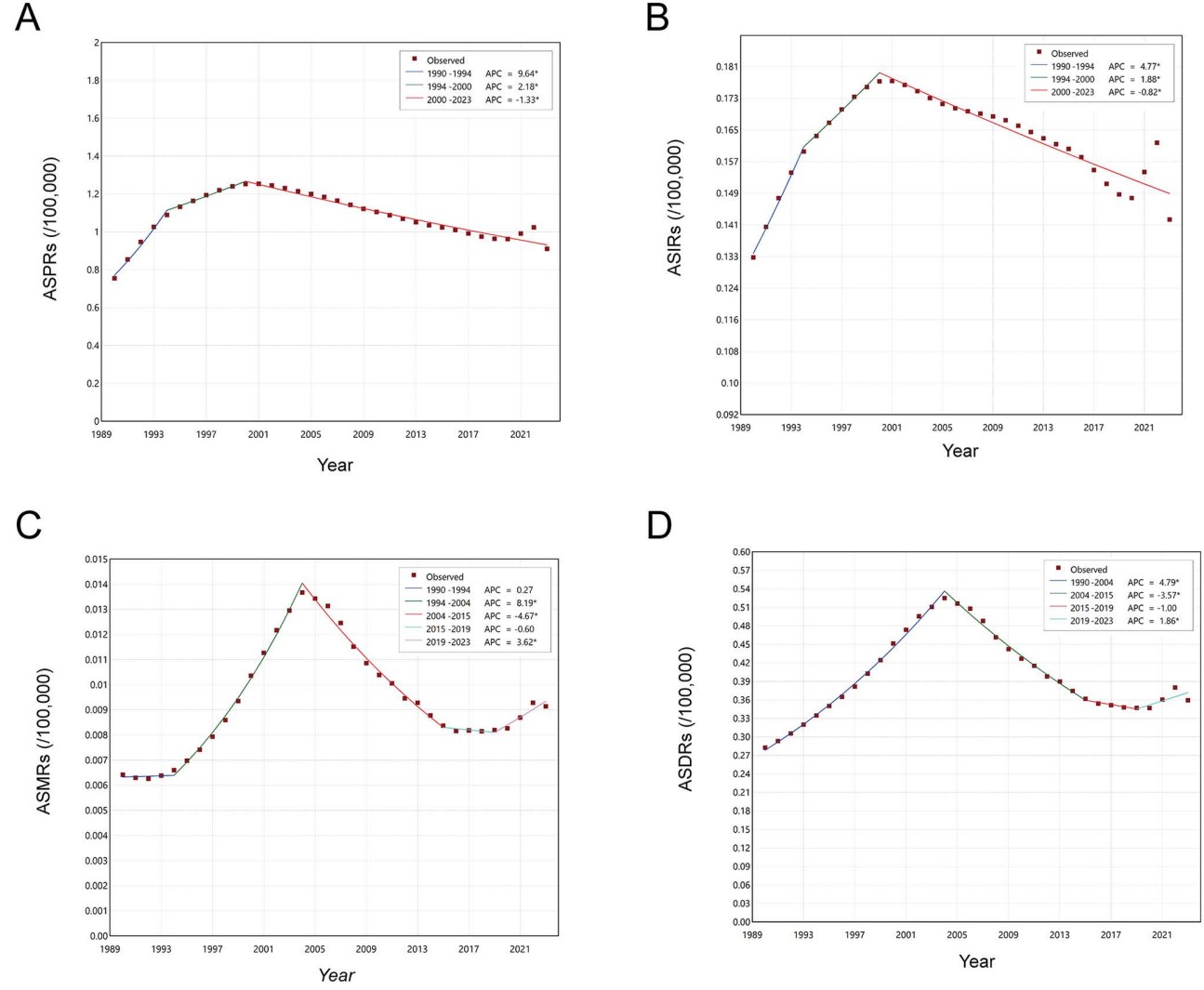

**Fig 2. Joinpoint regression of the disease burden of asbestosis in China.** The joinpoint regression of the age-standardized rates of prevalence **(A)**, incidence **(B)**, deaths **(C)**, and DALYs **(D)** for asbestosis. Abbreviations: ASPRs: age-standardized prevalence rates; ASIRs: age-standardized incidence rates; ASMRs: age-standardized mortality rates; DALYs: disability-adjusted life years; ASDRs: age-standardized DALY rates; APCs: annual percent changes.

## Evolution of asbestos-attributable cancers burden in China, 1990–2023

The absolute numbers of deaths and DALYs for mesothelioma, tracheal/bronchus/lung cancer, laryngeal cancer, and ovarian cancer attributable to asbestos exposure generally increased throughout the study period (Figs 3 and 4). ASMRs and ASDRs for mesothelioma declined before 2004, rose during 2004–2010, and then decreased steadily until 2020, followed by an increase after 2020 (Figs 3A and 4A, S2 Table). For tracheal/bronchus/lung cancer, the burden increased prior to 2009, with a particularly sharp rise between 2004 and 2009; a non-significant decline occurred during 2009–2017 (p > 0.05), followed by a largely stable period through 2020, after which the indicators rose again (Figs 3B and 4B). Laryngeal cancer demonstrated similar temporal patterns, with rates decreasing before 2004–2005, increasing during the mid-2000s, markedly declining after 2010–2011, and rising again from 2020 onward (Figs 3C and 4C). Ovarian cancer showed more complex trajectories with multiple joinpoints: the burden declined from 1990 to 2005, increased from 2005 to 2011,

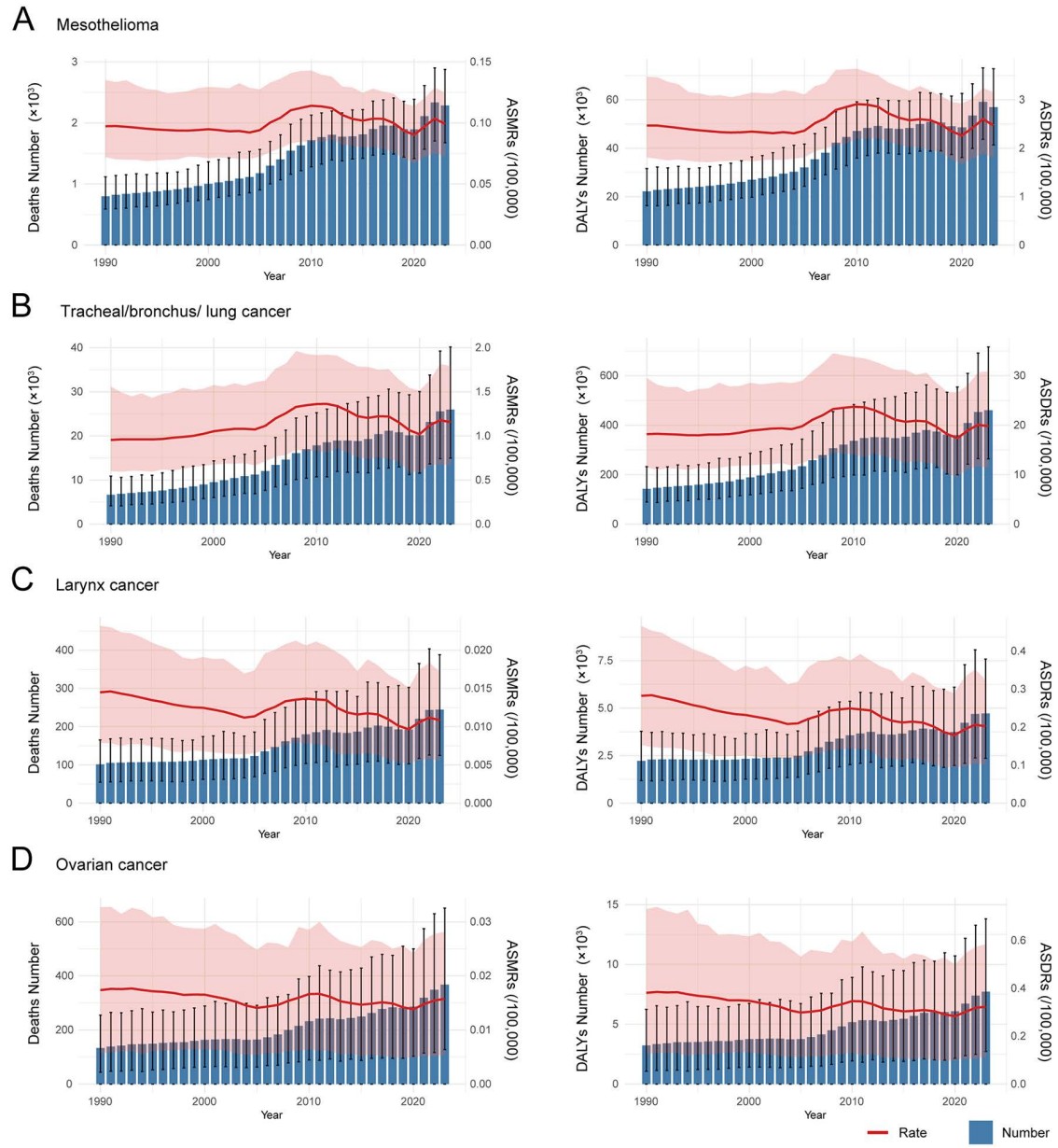

**Fig 3. Disease burden of asbestos-related cancers in China.** The number and age-standardized rates of deaths and DALYs for mesothelioma **(A)**, tracheal/bronchus/lung cancer **(B)**, laryngeal cancer **(C)**, and ovarian cancer **(D)**. Abbreviations: ASMRs: age-standardized mortality rates; DALYs: disability-adjusted life years; ASDRs: age-standardized DALY rates; APCs: annual percent changes.

declined again through 2014, and showed no significant changes from 2014 to 2020, before increasing after 2020 (Figs 3D and 4D).

## Sex Differences in the Disease Burden of ARDs

Across all ARDs, males consistently experienced substantially higher burdens than females (Figs 5 and 6). Although absolute levels differed, sex-specific trends in age-standardized rates were broadly similar. The higher male burden aligns with the predominance of men in physically demanding occupations with greater asbestos exposure [8,10].

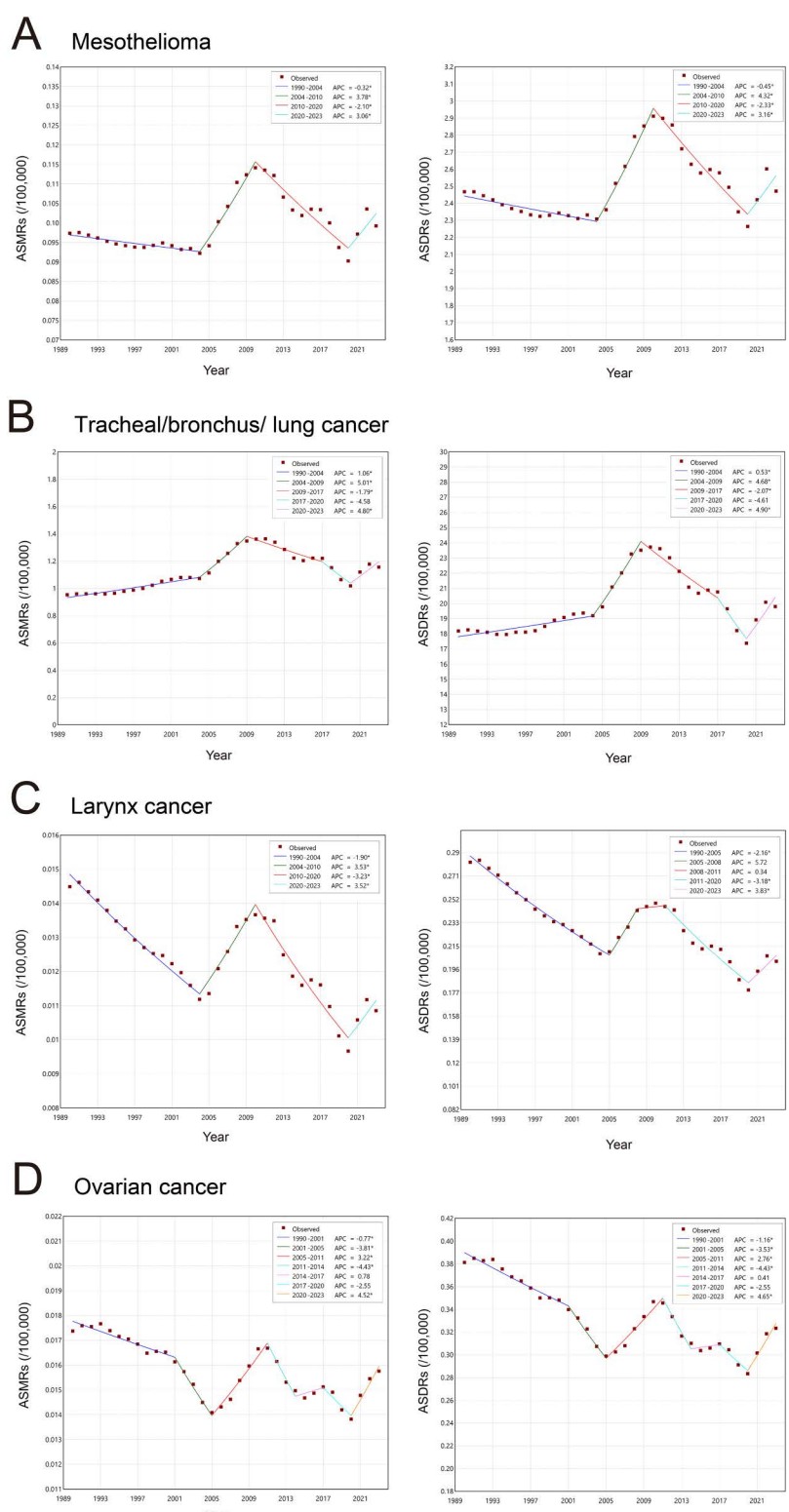

**Fig 4. Joinpoint regression of the disease burden of asbestos-related cancers in China.** The joinpoint regression of ASMRs and ASDRs for mesothelioma **(A)**, tracheal/bronchus/lung cancer **(B)**, laryngeal cancer **(C)**, and ovarian cancer **(D)**. Abbreviations: ASMRs: age-standardized mortality rates; DALYs: disability-adjusted life years; ASDRs: age-standardized DALY rates; APCs: annual percent changes.

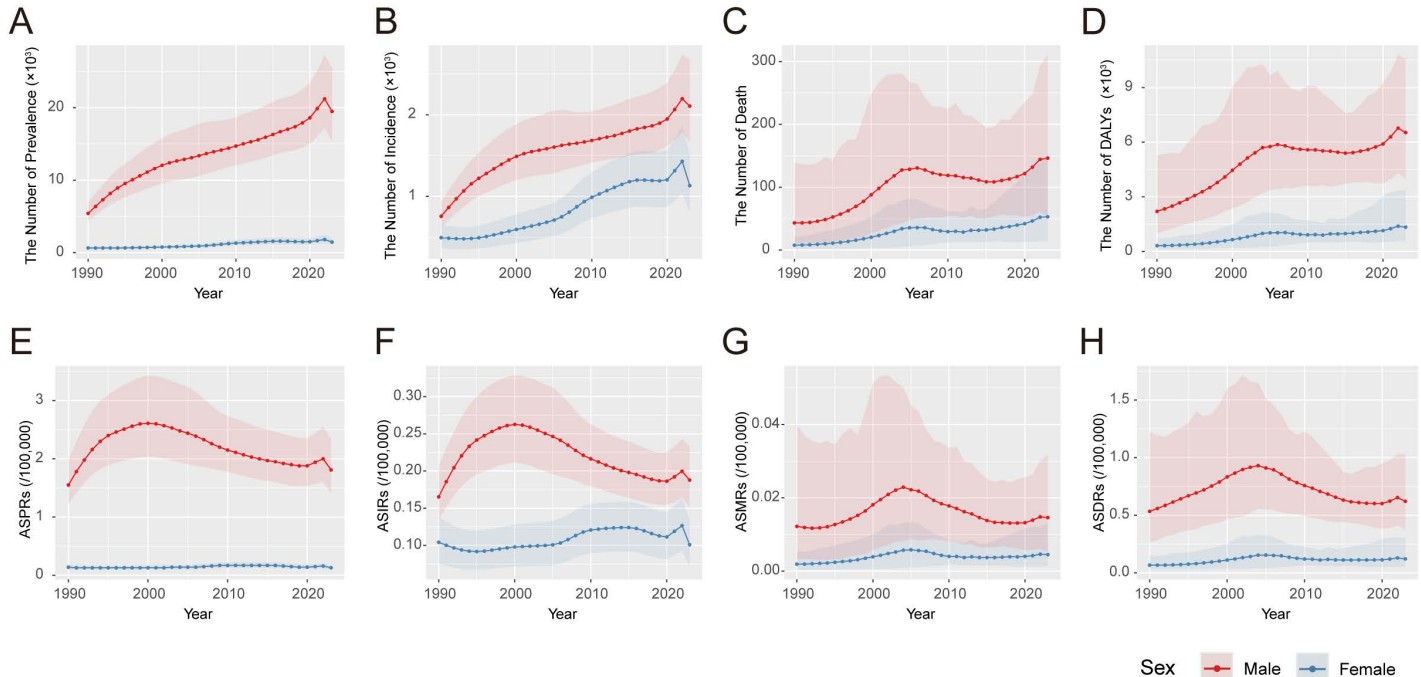

**Fig 5. Temporal trends in the disease burden of asbestosis in China by sex.** The number (A–D) and age-standardized rates (E–H) of prevalences, incidences, deaths, and DALYs for asbestosis in males and females. Abbreviations: ASPRs: age-standardized prevalence rates; ASIRs: age-standardized incidence rates; ASMRs: age-standardized mortality rates; DALYs: disability-adjusted life years; ASDRs: age-standardized DALY rates.

## Age-specific burden of ARDs

In 2023, no asbestosis cases were recorded among individuals aged ≤14 years (Fig 7). The number of prevalence (Fig 7A), incidence (Fig 7B), deaths (Fig 7C), and DALYs (Fig 7D) of asbestosis peaked at ages 65–74 in both sexes, while incidence peaked at 65–69 years in males and 55–59 years in females. ASPRs, ASMRs, and ASDRs of asbestosis increased progressively with age, more markedly in males (Figs 7E, G, H). ASIRs (Fig 7F) peaked at 70–74 years in males and 60–64 years in females, consistent with the distribution of incident cases.

For mesothelioma, deaths and DALYs peaked at 55–59 years in males and 65–74 years in females (Figs 8A-B). For tracheal/bronchus/lung and laryngeal cancers (Figs 8A–B), male deaths and DALYs peaked at 75–79 years, with a secondary peak at 55–59 years; female burdens peaked at older ages (70–84 years). For ovarian cancer (Figs 8A-B), deaths and DALYs peaked at 70–74 years. Among males, ASMRs and ASDRs for most cancers were highest at ages 85–94, with a distinct secondary peak at 55–59 years for mesothelioma (Figs 8C-D). Among females, ASMRs and ASDRs of asbestos-attributable cancers generally increased with age, except for mesothelioma and ovarian cancer, where ASDRs peaked at 70–74 years (Figs 8C-D).

### Comparison of Occupational Asbestos Exposure Regulations Between China and Developed Countries

China's regulations were less stringent than those in the United States, United Kingdom, Germany, and France (S3 Table). Unlike the full bans implemented in several developed countries, China and the U.S. had not fully prohibited asbestos use. China's 8-hour time-weighted average exposure limit (0.8 fibers/cm³) was substantially higher than limits in developed countries (0.01–0.1 fibers/cm³). Requirements for high-efficiency particulate air (HEPA) filtration, advanced personal

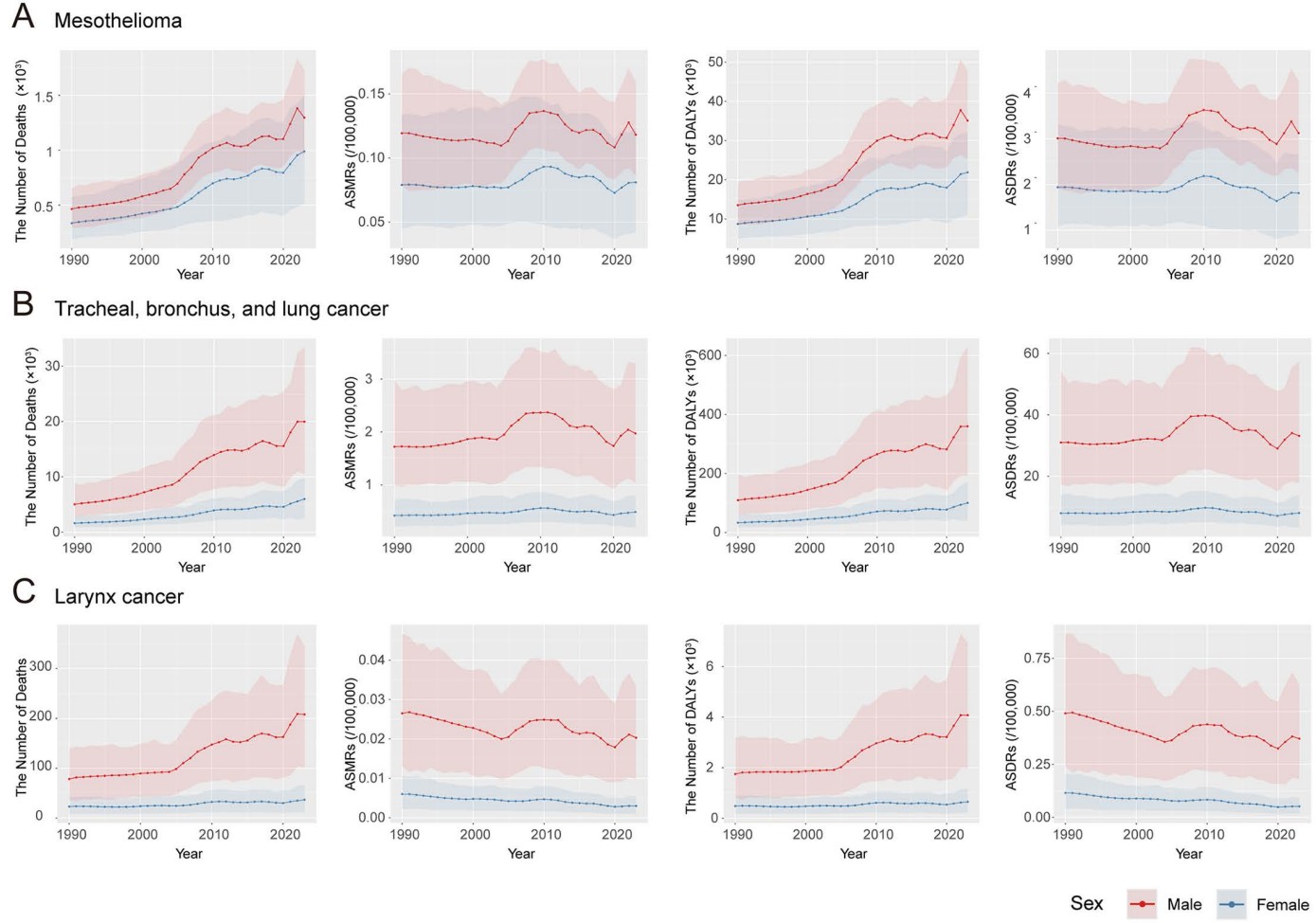

**Fig 6. Temporal trends in deaths and DALYs of asbestos-related cancers in China by sex.** The number and age-standardized rates of deaths and DALYs for mesothelioma **(A)**, tracheal/bronchus/lung cancer **(B)**, and laryngeal cancer **(C)** in males and females. Abbreviations: ASMRs: age-standardized mortality rates; DALYs: disability-adjusted life years; ASDRs: age-standardized DALY rates.

protective equipment, and mandated monitoring frequency were either absent or only principle-based in China, whereas other countries had clear legal specifications.

## Discussion

Over the past several decades, substantial global efforts have been made to eliminate asbestos-related diseases. However, to this day, these diseases continue to impose a significant burden on public health worldwide. Estimates from the WHO/ILO indicate that occupational exposure to asbestos is responsible for approximately 200,000 deaths and nearly 4 million DALYs worldwide each year [15]. According to previous GBD studies, from 1990 to 2017, both the absolute number of incident cases and the ASIRs of asbestosis continued to rise globally [16]. By 2019, global ASPRs and ASMRs of asbestosis had increased compared with 1990, while ASDRs had declined [11]. Mesothelioma is a rare but representative malignancy, over 90% of mesothelioma-related deaths were attributable to asbestos exposure [6]. Between 1990 and 2019, although global ASMRs and ASDRs for mesothelioma showed a gradual decline, absolute deaths and DALYs continued to increase [6]. Similar patterns were observed in asbestos-attributable tracheal/bronchus/lung cancer,

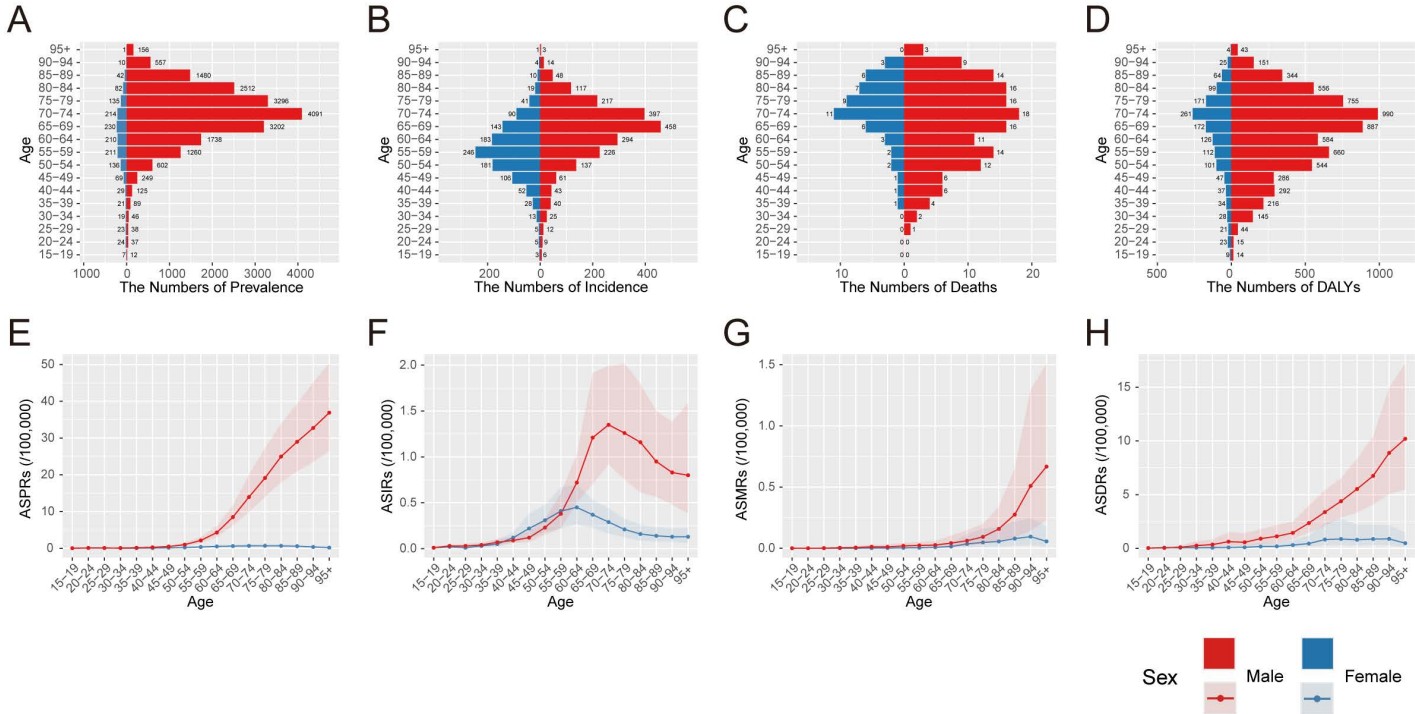

**Fig 7. Disease burden of asbestosis in China in 2023 by age group and sex.** The number (A–D) and age-standardized rates (E–H) of prevalence, incidence, deaths, and DALYs across different age groups in males and females. Abbreviations: ASPRs: age-standardized prevalence rates; ASIRs: age-standardized incidence rates; ASMRs: age-standardized mortality rates; DALYs: disability-adjusted life years; ASDRs: age-standardized DALY rates.

laryngeal cancer, and ovarian cancer, where absolute deaths and DALYs increased over time while age-standardized rates declined [17].

China has the largest occupational disease population in the world, making analysis of its disease burden critical for national and global efforts to eliminate occupational diseases. Previous studies have reported the disease burden of asbestosis [11] and asbestos-related mesothelioma [18] in China from 1990 to 2019 using data from the GBD study. However, updated estimates for 2020–2023 have not yet been published, and there are currently no dedicated reports on the burden of asbestos-attributable tracheal/bronchus/lung cancer, laryngeal cancer, or ovarian cancer in China. Our results show that, consistent with global patterns, the absolute burden of ARDs in China has generally increased over time, although rates have declined from their historical peaks. In detail, we analyzed burden trends using joinpoint analysis. Although turning points and trends varied across diseases, several similarities emerged. First, the peak burden of asbestosis has passed, with highest ASPRs and ASIRs occurring in 2000, and highest ASMRs and ASDRs in 2004. Second, for the four asbestos-attributable cancers, several overlapping key time points were identified: (1) 2004 or 2005, after which the disease burden of all four cancers increased rapidly; (2) around 2010, after which the disease burden of all four cancers began to decline; and (3) 2020, after which a modeled increase was observed. Notably, ASMRs and ASDRs of asbestosis also increased after 2020, all with statistical significance.

Among these results, particular attention should be paid to the increase in the burden of ARDs observed after 2020. Although most of these rates showed a decline again in 2023, this trend still deserves attention. First, considering that asbestos production and consumption in China have shown a declining or stable pattern since 2011 [18,19], the post-2020 increase in ARD burden may be associated with the long latency period of ARDs, which typically develop decades

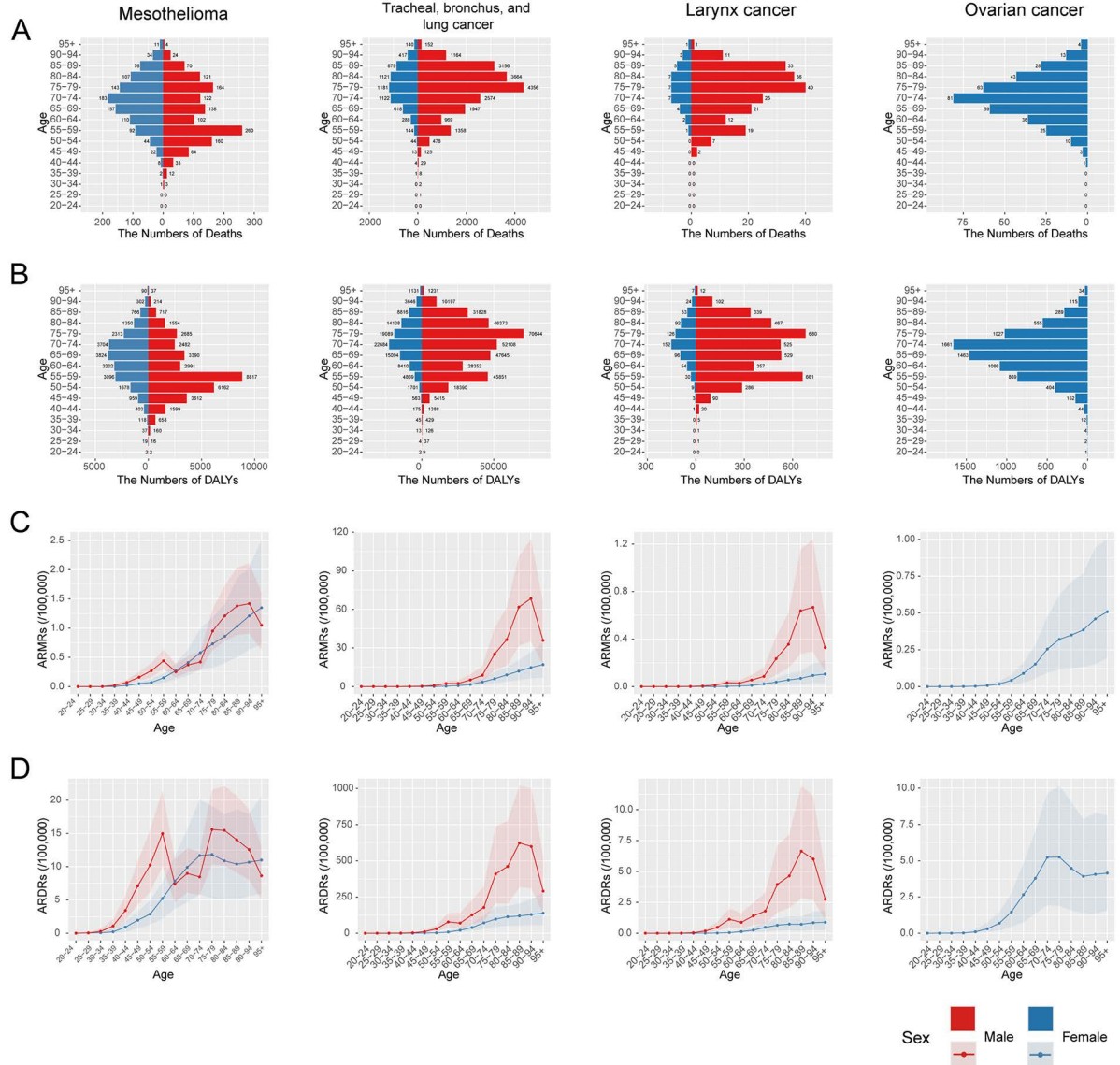

**Fig 8. Disease burden of asbestos- attributable cancers in China in 2023 by age group and sex.** The number (A, B) and age-standardized rates (C, D) of deaths and DALYs for mesothelioma, tracheal/bronchus/lung cancer, laryngeal cancer, and ovarian cancer across different age groups in males and females. Abbreviations: ASMRs: age-standardized mortality rates; DALYs: disability-adjusted life years; ASDRs: age-standardized DALY rates.

after exposure. Second, the COVID-19 pandemic may have influenced health-care utilization, disease diagnosis, death certification, and reporting systems, which could affect mortality and DALY estimates. Third, updates to the GBD modeling framework, input data sources, or cause-of-death redistribution algorithms may also influence recent estimates. It should be noted that all estimates from the GBD study are model-based estimates generated through standardized epidemiological modeling rather than direct surveillance data. Therefore, while the rebound in the burden of ARDs after 2020 should be interpreted with caution and requires epidemiological validation, the possibility that it may represent an early warning signal should not be overlooked.

Furthermore, we analyzed the differences in disease burden by sex and age group. The results clearly showed that males consistently bore a significantly higher burden of asbestos-related diseases than females across years and age groups. While the peak DALY rates for both sexes were generally observed between the ages of 70–79, most asbestos-related diseases in males showed a secondary DALY peak at 50–59 years, suggesting that disease onset in men may occur earlier than in women. First, this finding is consistent with previous studies, which have shown that asbestos-related diseases tend to be more severe in males than in females. This is largely attributable to the predominance of males in physically demanding occupations, where higher levels of asbestos exposure are more likely [17]. Additionally, co-exposure to other risk factors such as smoking may further exacerbate disease severity in men [17]. Second, and more likely, the secondary peak observed at ages 55–59 years in males may reflect occupational exposure patterns and birth-cohort effects. Individuals in this age group were likely exposed to asbestos during periods of rapid industrial expansion in the 1980s and 1990s, when occupational protection measures were relatively limited. Considering the typical latency period of mesothelioma (20–40 years), this exposure period corresponds well with disease occurrence in middle-to-late adulthood. This pattern may therefore represent a cohort effect rather than an age effect alone.

Given the lack of effective treatments [4], reducing asbestos exposure and banning its use remain the most fundamental strategies for preventing related diseases. Thus, stringent labor protection laws and occupational safety standards are essential for mitigating the ongoing burden of asbestos-related diseases. However, China's legal framework for asbestos control still lags considerably behind that of countries such as the U.S., France, Germany, and the U.K. These findings highlight the public health importance of asbestos exposure control and support the need for strengthened occupational health policies and exposure monitoring systems in China. It should be emphasized that, as this study does not directly evaluate regulatory enforcement, exposure intensity, or compliance, it is not intended to assess causality between past health policies and asbestos exposure, but rather to inform future policy and research.

Despite its strengths, this GBD-based analysis has several inherent limitations. First, the GBD database integrates data from diverse sources, and many estimates rely on modeling rather than direct surveillance [8]. As a result, discrepancies may exist between GBD estimates and the true epidemiological situation. Second, the GBD framework lacks individual-level data on asbestos exposure intensity, duration, or co-exposures. Cancer attribution relies on previously published relative risk estimates and PAFs, which limits the ability to adjust for important confounders—such as smoking, air pollution, and occupational co-exposures—that vary across regions and time periods. Additionally, the synergistic interaction between asbestos and smoking in lung cancer cannot be fully captured. Therefore, the estimated asbestos-attributable burden may be subject to residual confounding and should be interpreted with caution. Third, GBD estimates depend heavily on the quality and completeness of input data. Underreporting of occupational diseases remains common, and asbestos exposure outside industrial settings (e.g., environmental or demolition-related exposure) is even more difficult to capture [20]. These source-level limitations may propagate through the modeling framework, potentially leading to underestimation of the true disease burden. Overall, these limitations underscore the need for more direct, comprehensive occupational exposure monitoring systems and population-based epidemiological data to improve future assessments.

## Conclusions

Although China has made measurable progress in reducing the burden of asbestos-related diseases, our findings indicate that the recent modeled increase in age-standardized mortality and disability rates after 2020 may represent a concerning reversal of earlier trends, warranting further investigation. The findings support the need for continued evaluation of labor protection laws and asbestos control standards in China to further reduce the burden of asbestos-related diseases.

## Supporting information

**S1 Table. Joinpoint regression of asbestosis in China, 1990–2023.**
(DOCX)

**S2 Table. Joinpoint regression of asbestos-related cancers in China, 1990–2023.**
(DOCX)

**S3 Table. Legal Regulations on Protective Requirements for Asbestos Operations in China and Major Industrialized Countries in Europe and North America.**
(DOCX)

## Author contributions

**Conceptualization:** Duo An.

**Data curation:** Zherui Shi, Chenxiao Zhang.

**Project administration:** Hanchao Li.

**Software:** Shichang Song, Lu Wang.

**Visualization:** Shichang Song, Lu Wang.

**Writing – original draft:** Duo An.

**Writing – review & editing:** Zherui Shi, Chenxiao Zhang, Hanchao Li.

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
