## [Decision Letter · Decision Letter 0]

12 Mar 2026

PONE-D-25-67182Disease Burden of Asbestos-Related Diseases in China (1990–2023) Based on GBD Estimates: A Call for Stronger Labor Protection LawsPLOS One

Dear Dr. Li,

Thank you for submitting your manuscript to PLOS ONE. After careful consideration, we feel that it has merit but does not fully meet PLOS ONE’s publication criteria as it currently stands. Therefore, we invite you to submit a revised version of the manuscript that addresses the points raised during the review process.

We look forward to receiving your revised manuscript.

Kind regards,

Xingyu Xiong, Ph.D.

Academic Editor

PLOS One

Journal Requirements:

2. In the online submission form you indicate that your data is not available for proprietary reasons and have provided a contact point for accessing this data. Please note that your current contact point is a co-author on this manuscript. According to our Data Policy, the contact point must not be an author on the manuscript and must be an institutional contact, ideally not an individual. Please revise your data statement to a non-author institutional point of contact, such as a data access or ethics committee, and send this to us via return email. Please also include contact information for the third party organization, and please include the full citation of where the data can be found.

4. Please ensure that you refer to Figure 5 in your text as, if accepted, production will need this reference to link the reader to the figure.

Additional Editor Comments:

Thanks for submitting your work to PLOS ONE. Your manuscript has now been assessed by our editorial team and external peer experts. While they found it interesting, you will see that they have raised many serious problems and are advising that you revise your manuscript thoroughly. At the same time, please submit the point-by-point responses to reviewers' comments. If you are prepared to undertake the work required, I would be pleased to reconsider my decision. Please note that this revision decision does not assure the acceptance of your work. Thanks for the opportunity to consider your work.

Reviewers' comments:

Reviewer's Responses to Questions

**Comments to the Author**

1. Is the manuscript technically sound, and do the data support the conclusions?

Reviewer #1: Yes

Reviewer #2: Yes

Reviewer #3: Yes

2. Has the statistical analysis been performed appropriately and rigorously? 

Reviewer #1: Yes

Reviewer #2: Yes

Reviewer #3: Yes

3. Have the authors made all data underlying the findings in their manuscript fully available?

Reviewer #1: Yes

Reviewer #2: No

Reviewer #3: Yes

4. Is the manuscript presented in an intelligible fashion and written in standard English?

Reviewer #1: No

Reviewer #2: Yes

Reviewer #3: Yes

5. Review Comments to the Author

Reviewer #1: The article "Disease Burden of Asbestos-Related Diseases in China (1990–2023) Based on GBD Estimates: A Call for Stronger Labor Protection Laws" presents the results of a descriptive study of asbestos-related diseases. For clarity and to facilitate evaluation, I suggest highlighting the study's objectives. I believe you could choose between two expressions found in the introduction and phrase them as the objectives: "Updated analyses are essential for understanding current patterns, guiding prevention strategies, and informing occupational health policy," or "These findings aim to provide evidence for clinicians, epidemiologists, and policymakers to strengthen occupational disease prevention and improve asbestos-control policies in China." The objectives should also be included in the abstract.

Reviewer #2: Dear Authors,

Thank you for the opportunity to review your manuscript, “Disease Burden of Asbestos-Related Diseases in China (1990–2023) Based on GBD Estimates: A Call for Stronger Labor Protection Laws.” This is a timely and policy-relevant study addressing an important occupational and environmental health issue. Using updated Global Burden of Disease (GBD) 2023 data and Joinpoint regression, the manuscript provides a comprehensive assessment of long-term trends in asbestosis and asbestos-attributable cancers in China.

The manuscript is generally well-structured and clearly written. The use of age-standardized rates, sex-stratified analyses, and long-term trend modeling strengthens the contribution. The topic is highly relevant to occupational health policy and public health planning.

I commend you for:

• Addressing a major occupational health burden in a country with historically high asbestos production and use.

• Utilizing updated GBD 2023 data to provide current national-level estimates.

• Applying Joinpoint regression to identify meaningful temporal inflection points.

• Including sex- and age-specific analyses to better characterize vulnerable populations.

• Linking epidemiological findings to occupational health policy implications.

I believe the following revisions would strengthen the manuscript:

1. Clarify Methodological Transparency and Reproducibility

• Specify the exact GBD iteration/version and data download date used in the analysis.

• Clarify whether uncertainty intervals (95% UI) from GBD were incorporated into Joinpoint analyses or whether point estimates only were used.

• Describe the criteria for selecting the maximum number of joinpoints (e.g., permutation test, BIC).

• Indicate whether alternative model specifications or sensitivity analyses were conducted.

• Consider providing supplementary material detailing model specifications or R scripts to improve reproducibility.

2. Refine Interpretation of the Post-2020 “Rebound”

• The increase observed from 2020–2023 spans a short time window; please perform or discuss a sensitivity comparison between pre-2020 projected trends and observed post-2020 data.

• Discuss possible COVID-19-related data distortions, mortality coding changes, or modeling artifacts that may influence the apparent rebound.

• Avoid strong causal language implying regulatory insufficiency unless directly supported by additional evidence.

3. Clarify Cancer Attribution Methodology

• Provide a concise explanation of how GBD calculates asbestos-attributable burden (comparative risk assessment framework and population attributable fractions).

• Expand discussion of potential residual confounding (e.g., smoking for lung and laryngeal cancer).

• Emphasize the ecological nature of GBD modeling and its limitations for causal inference.

4. Moderate Policy Conclusions

• The manuscript does not directly evaluate regulatory enforcement, exposure intensity, compliance, or surveillance systems.

• Reframe policy implications as hypothesis-generating rather than definitive conclusions.

• Clarify that the study demonstrates descriptive burden trends rather than direct evidence of regulatory effectiveness.

5. Strengthen Age-Specific Interpretation

• Expand discussion of biological and occupational explanations for the secondary mesothelioma peak at 55–59 years in males.

• Consider discussing potential birth-cohort effects and latency-period differences.

Minor Edits

• Correct typographical errors (e.g., “Fugure” instead of “Figure”).

• Ensure consistent capitalization of terms such as “asbestos-attributable cancer.”

• Revise the Data Availability statement to clearly indicate that data are publicly available from GHDx with a direct URL.

• Clarify the rationale for ethics exemption given the use of publicly available secondary data.

• Simplify Tables 1 and 2 where possible and reduce excessive decimal precision in APC values.

• Improve readability of figure axis labels and harmonize color schemes across sex-stratified figures.

Overall, this is an important and well-executed descriptive epidemiological study with strong public health relevance. With the revisions outlined above, particularly regarding methodological clarity, cautious interpretation of post-2020 trends, and moderation of policy conclusions, the manuscript will make a meaningful contribution to the literature on asbestos-related disease burden and occupational health policy in China.

Best wishes,

Reviewer

Reviewer #3: This is a strong, data-rich, policy-relevant GBD-based trend analysis. The topic is appropriate for PLOS ONE, the methods are standard and reproducible, and the manuscript is generally well written. The long-time span (1990–2023) and the post-2020 rebound finding are clear strengths.

That said, the paper would benefit from methodological clarifications, tighter framing of novelty, and careful handling of causal/policy claims; especially since it relies entirely on modeled GBD estimates.

Major strengths of the paper include timeliness & novelty, which extends previous China analyses beyond 2019 to 2023, which is genuinely new. The post-2020 rebound is interesting and worth highlighting. In addition to that, the manuscript’s comprehensive scope, appropriate methodology, clear demographic stratification, and policy relevance makes it high-value output.

Below are my comments and suggestion that need to be addressed.

1. Over-interpretation of GBD Estimates

This is the biggest issue. The manuscript sometimes reads as if these are observed epidemiological data, not modeled estimates. Phrases like “temporary rebound occurred” risk implying causality.

My suggestion is that the authors should add clearer language in Methods and Discussion emphasizing that these are modeled estimates, not surveillance data; and Joinpoints indicate statistical trend changes, not real-world events

Example fix: “A statistically modeled increase was observed after 2020…”

2. Causal Language vs Descriptive Design

This is a descriptive trend analysis, yet the paper occasionally implies causation, such as “suggesting current asbestos control policies may be insufficient” and “underscores an urgent need for stricter bans”

The authors need to reframe to: “These findings support the need for…” and “These patterns are consistent with insufficient control…”

This keeps it aligned with PLOS ONE’s non-causal, evidence-neutral stance.

3. Justification for Including Ovarian Cancer

Ovarian cancer is controversial in asbestos literature. The GBD attribution is valid, but readers may question it. The rationale is only lightly mentioned.

I suggest adding 1–2 sentences in the Introduction or Methods clearly stating that ovarian cancer is included per GBD risk attribution framework, and cite WHO/ILO or GBD methodology papers more clearly.

4. Post-2020 Rebound: Competing Explanations Missing

The rebound after 2020 is interesting—but currently underexplained. Possible contributors not discussed include COVID-19-related diagnostic delays, death certification artifacts, model recalibration in GBD 2023, lagged mortality reporting

My recommendation is that, add a short paragraph acknowledging alternative explanations.

Below are minor Issues & Technical Points

= A few typos: “Fugure” => Figure (multiple times), “ASDYs” => ASDRs (line ~209).

= Some sentences are long and could be tightened in the Discussion.

= Tables 1 & 2 are dense but acceptable. Consider adding one summary figure highlighting all ARDs post-2020 trends, Or a small schematic timeline of key joinpoints

= The Discussion is well-referenced, logically structured, and consistent with results. But it could be sharper by separating global context vs China-specific interpretation, reducing repetition around latency explanation, and being more cautious with policy enforcement claims

6. PLOS authors have the option to publish the peer review history of their article (what does this mean?). If published, this will include your full peer review and any attached files.

Reviewer #1: No

Reviewer #2: No

Reviewer #3: No

---

## [Author Response · Author response to Decision Letter 1]

28 Mar 2026

To Reviewer #1:

The article "Disease Burden of Asbestos-Related Diseases in China (1990–2023) Based on GBD Estimates: A Call for Stronger Labor Protection Laws" presents the results of a descriptive study of asbestos-related diseases. For clarity and to facilitate evaluation, I suggest highlighting the study's objectives. I believe you could choose between two expressions found in the introduction and phrase them as the objectives: "Updated analyses are essential for understanding current patterns, guiding prevention strategies, and informing occupational health policy," or "These findings aim to provide evidence for clinicians, epidemiologists, and policymakers to strengthen occupational disease prevention and improve asbestos-control policies in China." The objectives should also be included in the abstract.

Response: Thank you very much for your valuable comments on our manuscript. We fully agree that clearly stating the research objectives in both the introduction and abstract is essential for improving the paper's clarity and logical flow.

Following your suggestion, we have revised the manuscript accordingly. In the introduction, we have highlighted the study objective using your recommended phrasing: "Updated analyses are essential for understanding current patterns, guiding prevention strategies, and informing occupational health policy." We have also incorporated this objective into the abstract to ensure consistency throughout the paper.

We greatly appreciate your thorough review and constructive guidance, which have helped us strengthen our work.

To reviewer 2#:

Thank you for the opportunity to review your manuscript, “Disease Burden of Asbestos-Related Diseases in China (1990–2023) Based on GBD Estimates: A Call for Stronger Labor Protection Laws.” This is a timely and policy-relevant study addressing an important occupational and environmental health issue. Using updated Global Burden of Disease (GBD) 2023 data and Joinpoint regression, the manuscript provides a comprehensive assessment of long-term trends in asbestosis and asbestos-attributable cancers in China.

The manuscript is generally well-structured and clearly written. The use of age-standardized rates, sex-stratified analyses, and long-term trend modeling strengthens the contribution. The topic is highly relevant to occupational health policy and public health planning.

I commend you for:

• Addressing a major occupational health burden in a country with historically high asbestos production and use.

• Utilizing updated GBD 2023 data to provide current national-level estimates.

• Applying Joinpoint regression to identify meaningful temporal inflection points.

• Including sex- and age-specific analyses to better characterize vulnerable populations.

• Linking epidemiological findings to occupational health policy implications.

I believe the following revisions would strengthen the manuscript:

1. Clarify Methodological Transparency and Reproducibility

• Specify the exact GBD iteration/version and data download date used in the analysis.

• Clarify whether uncertainty intervals (95% UI) from GBD were incorporated into Joinpoint analyses or whether point estimates only were used.

• Describe the criteria for selecting the maximum number of joinpoints (e.g., permutation test, BIC).

• Indicate whether alternative model specifications or sensitivity analyses were conducted.

• Consider providing supplementary material detailing model specifications or R scripts to improve reproducibility.

Response: We sincerely appreciate your thoughtful and detailed review of our manuscript. Your suggestions have been very helpful in guiding our revisions.

The last access date of the data is July 29, 2025, added on line 85, page 5 of the manuscript.

Point estimates were used, but they were not used alone; the GBD uncertainty intervals (via standard errors) were simultaneously incorporated into the analysis by weighting the data points based on their precision. Specifically, the age-standardized rates (point estimates) served as the dependent variable. Under the "Heteroscedastic/Correlated Errors Option," we specified "Standard Error (Provided)" and designated the corresponding standard error column as the error term, which allowed the model to weight each observation inversely to its variance. This approach ensures that years with larger uncertainty (higher standard errors) contribute less to the model fitting, while years with more precise estimates (smaller standard errors) exert greater influence. Thus, although point estimates were used as input values, the uncertainty from GBD was directly incorporated into the trend analysis.

The selection criteria involved setting a maximum number of joinpoints and then using a data-driven method to determine the final model. Specifically, the maximum number of joinpoints was set to six, defining the upper limit of model complexity to be explored. The final model was then selected using the Bayesian Information Criterion (BIC), specifically the Weighted BIC method under "Data Driven BIC Methods." This approach automatically identifies the model with the optimal balance between goodness-of-fit and parsimony by comparing BIC values across models with 0 to 6 joinpoints, thereby avoiding overfitting.

Alternative model specifications or sensitivity analyses were not conducted. The Joinpoint software employs a well-established model selection framework (Weighted BIC) that automatically balances goodness-of-fit and parsimony, and we considered this approach appropriate for identifying the optimal trend structure. Additionally, the inclusion of standard errors as weights in the model already accounts for the varying precision of the input data, thereby enhancing the reliability of the estimates.

All details regarding the Joinpoint analysis have been supplemented on page 7, line 120 to 125 of the manuscript.

To enhance data transparency and reproducibility, all R code and data used in this analysis have been made publicly available on figshare at https://doi.org/10.6084/m9.figshare.31745317. This information has been added on page 7, line 135 to 140 of the manuscript.

2. Refine Interpretation of the Post-2020 “Rebound”

• The increase observed from 2020–2023 spans a short time window; please perform or discuss a sensitivity comparison between pre-2020 projected trends and observed post-2020 data.

• Discuss possible COVID-19-related data distortions, mortality coding changes, or modeling artifacts that may influence the apparent rebound.

• Avoid strong causal language implying regulatory insufficiency unless directly supported by additional evidence.

Response: We agree that the increase observed during 2020–2023 spans a relatively short time window and should be interpreted cautiously. We have revised the manuscript to avoid strong causal language and clarified that the trends identified in this study are based on modeled estimates from the Global Burden of Disease (GBD) study rather than direct surveillance data. We also added a discussion addressing alternative explanations for the post-2020 increase, including COVID-19-related disruptions in healthcare utilization and mortality reporting, possible mortality coding changes, delayed diagnosis, and potential updates to the GBD modeling framework. We have revised the relevant sections in the Discussion accordingly (line 289 to 291, page 14).

3. Clarify Cancer Attribution Methodology

• Provide a concise explanation of how GBD calculates asbestos-attributable burden (comparative risk assessment framework and population attributable fractions).

• Expand discussion of potential residual confounding (e.g., smoking for lung and laryngeal cancer).

• Emphasize the ecological nature of GBD modeling and its limitations for causal inference.

Response: Thank you for this suggestion. We have added a concise explanation in the Methods section describing how the GBD study estimates asbestos-attributable cancer burden using the comparative risk assessment framework and population attributable fractions (line 89-94, page 5). We also expanded the Discussion to address potential residual confounding factors, particularly smoking for lung and laryngeal cancer, and emphasized the ecological and model-based nature of GBD estimates and their limitations for causal inference (line 331-343, page 16) .

4. Moderate Policy Conclusions

• The manuscript does not directly evaluate regulatory enforcement, exposure intensity, compliance, or surveillance systems.

• Reframe policy implications as hypothesis-generating rather than definitive conclusions.

• Clarify that the study demonstrates descriptive burden trends rather than direct evidence of regulatory effectiveness.

Response: We agree with the reviewer that this study does not directly evaluate regulatory enforcement, exposure intensity, compliance, or surveillance systems. Therefore, we revised the Discussion and Conclusions to moderate policy-related language and reframed the policy implications as descriptive evidence that may inform future occupational health policy rather than definitive conclusions regarding regulatory effectiveness (line 322 to 327, page 16).

5. Strengthen Age-Specific Interpretation

• Expand discussion of biological and occupational explanations for the secondary mesothelioma peak at 55–59 years in males.

• Consider discussing potential birth-cohort effects and latency-period differences.

Response: Thank you for this helpful suggestion. We expanded the Discussion to provide additional biological and occupational explanations for the secondary mesothelioma peak at ages 55–59 years in males, including latency period considerations, occupational exposure patterns, and potential birth-cohort effects associated with industrial expansion periods in China (line 309-316, page 15).

Minor Edits

• Correct typographical errors (e.g., “Fugure” instead of “Figure”).

• Ensure consistent capitalization of terms such as “asbestos-attributable cancer.”

• Revise the Data Availability statement to clearly indicate that data are publicly available from GHDx with a direct URL.

• Clarify the rationale for ethics exemption given the use of publicly available secondary data.

• Simplify Tables 1 and 2 where possible and reduce excessive decimal precision in APC values.

• Improve readability of figure axis labels and harmonize color schemes across sex-stratified figures.

Overall, this is an important and well-executed descriptive epidemiological study with strong public health relevance. With the revisions outlined above, particularly regarding methodological clarity, cautious interpretation of post-2020 trends, and moderation of policy conclusions, the manuscript will make a meaningful contribution to the literature on asbestos-related disease burden and occupational health policy in China.

Response: We have corrected typographical errors, standardized terminology throughout the manuscript, revised the Data Availability statement to clearly indicate that data are publicly available from the Global Health Data Exchange (GHDx), clarified the ethics statement regarding the use of publicly available secondary data, simplified tables by reducing excessive decimal precision in APC values, and improved figure readability and consistency across sex-stratified figures. In addition, we have split the original Figure 1 and Figure 2 so that the APC values are now presented directly in the Joinpoint plots. Accordingly, the original Table 1 and Table 2 have been moved to the supplementary materials, which enhances the clarity of the results presentation.

We are grateful to the reviewer for the valuable time and expertise invested in reviewing our manuscript. The insightful suggestions have substantially strengthened our study, and we appreciate the thoughtful guidance throughout the review process.

To reviewer 3#:

This is a strong, data-rich, policy-relevant GBD-based trend analysis. The topic is appropriate for PLOS ONE, the methods are standard and reproducible, and the manuscript is generally well written. The long-time span (1990–2023) and the post-2020 rebound finding are clear strengths.

That said, the paper would benefit from methodological clarifications, tighter framing of novelty, and careful handling of causal/policy claims; especially since it relies entirely on modeled GBD estimates.

Major strengths of the paper include timeliness & novelty, which extends previous China analyses beyond 2019 to 2023, which is genuinely new. The post-2020 rebound is interesting and worth highlighting. In addition to that, the manuscript’s comprehensive scope, appropriate methodology, clear demographic stratification, and policy relevance makes it high-value output.

Below are my comments and suggestion that need to be addressed.

1. Over-interpretation of GBD Estimates

This is the biggest issue. The manuscript sometimes reads as if these are observed epidemiological data, not modeled estimates. Phrases like “temporary rebound occurred” risk implying causality.

My suggestion is that the authors should add clearer language in Methods and Discussion emphasizing that these are modeled estimates, not surveillance data; and Joinpoints indicate statistical trend changes, not real-world events

Example fix: “A statistically modeled increase was observed after 2020…”

Response: We thank the reviewer for raising this critical point. We fully agree that it is essential to clearly distinguish between modeled GBD estimates and observed epidemiological data, and to avoid language that may inadvertently imply causality.

In response to this comment, we have made the following revisions to the manuscript:

In the Methods section, we added explicit language clarifying that all estimates used in this study are derived from the GBD modeling framework and represent modeled estimates rather than directly observed surveillance data. Specifically, we added: "The GBD study integrates data from vital registration systems, cancer registries, surveys, and modeling strategies to produce comparable estimates..." and emphasized that these are model-based outputs (line 86 to 88, page 5).

In the Abstract and Results sections, we carefully revised phrasing that could imply causality or real-world events. For example, we changed "a temporary rebound occurred" to "a statistically modeled increase was observed after 2020" to accurately reflect the nature of the underlying data and analytical methods (line 33 and 38, page 4).

In the Discussion section, we added a dedicated statement emphasizing that Joinpoint regression identifies statistical inflection points in modeled trends, which should not be interpreted as direct evidence of real-world events without complementary evidence. The added text reads: "It should be noted that all estimates from the GBD study are model-based estimates generated through standardized epidemiological modeling rather than direct surveillance data. Therefore, the trends identified in this study represent statistically modeled temporal patterns rather than direct observations of real-world incidence or mortality changes." (line 293 to 315, page 17)

We believe these revisions address the reviewer's concern and significantly improve the clarity and interpretative caution of the manuscript.

2. Causal Language vs Descriptive Design

This is a descriptive trend analysis, yet the paper occasionally implies causation, such as “suggesting current asbestos control policies may be insufficient” and “underscores an urgent need for stricter bans”

The authors need to reframe to: “These findings support the need for…” and “These patterns are consistent with insufficient control…”

This keeps it aligned with PLOS ONE’s non-causal, evidence-neutral stance.

Response: We thank the reviewer for this important observation. We agree that as a descriptive trend analysis, our study should avoid language that implies causation. Accordingly, we have removed phrases such as "suggesting current asbestos control policies may be insufficient" and "underscores an urgent need for stricter bans" (page 16, lines 322–327). Regarding asbestos policy, we now simply highlight the relative weaknesses of China's asbestos control policies compared with those of European and North American countries, as wel

---

## [Decision Letter · Decision Letter 1]

30 Apr 2026

Disease Burden of Asbestos-Related Diseases in China (1990–2023) Based on GBD Estimates: A Call for Stronger Labor Protection Laws

PONE-D-25-67182R1

Dear Dr. Li,

We’re pleased to inform you that your manuscript has been judged scientifically suitable for publication and will be formally accepted for publication once it meets all outstanding technical requirements.

Kind regards,

Xingyu Xiong, Ph.D.

Academic Editor

PLOS One

Additional Editor Comments (optional):

Reviewers' comments:

Reviewer's Responses to Questions

**Comments to the Author**

1. If the authors have adequately addressed your comments raised in a previous round of review and you feel that this manuscript is now acceptable for publication, you may indicate that here to bypass the “Comments to the Author” section, enter your conflict of interest statement in the “Confidential to Editor” section, and submit your "Accept" recommendation.

Reviewer #1: All comments have been addressed

Reviewer #3: All comments have been addressed

2. Is the manuscript technically sound, and do the data support the conclusions?

Reviewer #1: Yes

Reviewer #3: Yes

3. Has the statistical analysis been performed appropriately and rigorously? 

Reviewer #1: Yes

Reviewer #3: Yes

4. Have the authors made all data underlying the findings in their manuscript fully available?

Reviewer #1: Yes

Reviewer #3: Yes

5. Is the manuscript presented in an intelligible fashion and written in standard English?

Reviewer #1: Yes

Reviewer #3: Yes

6. Review Comments to the Author

Reviewer #1: The suggestions made have been incorporated, for which I thank the authors of the article “Disease Burden of Asbestos-Related Diseases in China (1990–2023) Based on GBD Estimates: A Call for Stronger Labor Protection Laws.” I recommend its publication.

Reviewer #3: I recommend acceptance of the manuscript.

The authors have satisfactorily addressed all concerns raised in my previous review. In particular, the manuscript now more clearly acknowledges that findings are based on modeled GBD estimates rather than direct surveillance data, causal language has been appropriately moderated, the inclusion of ovarian cancer has been justified, and alternative explanations for the post-2020 increase have been discussed.

Great work! Congratulations to the authors!

7. PLOS authors have the option to publish the peer review history of their article (what does this mean?). If published, this will include your full peer review and any attached files.

Reviewer #1: No

Reviewer #3: No

---

## [Editor Report · Acceptance letter]

PONE-D-25-67182R1

PLOS One

Dear Dr. Li,

I'm pleased to inform you that your manuscript has been deemed suitable for publication in PLOS One. Congratulations! Your manuscript is now being handed over to our production team.

Kind regards,

on behalf of

Dr. Xingyu Xiong

Academic Editor

PLOS One